# School Energy Consumption and Children's Obesity: Evidence from China

**Shangrong Han [1], Bo Han [1,*], Yan Zhu [2], Xiaojie Liu [1] and Limin Fu [3]**

[1] School of Statistics, Beijing Normal University, Beijing 100875, China
[2] College of Physical Education and Sports, Beijing Normal University, Beijing 100875, China; zhy0009@163.com
[3] Department of Sport Arts, Hebei Sport University, Shijiazhuang 050041, China; fulimin@hepec.edu.cn
[*] Correspondence: 202031011006@mail.bnu.edu.cn; Tel.: +86-181-1740-8750

**Abstract:** Rising obesity rates may lead to an increase in greenhouse gas emissions, undermining carbon neutrality goals. However, evidence of the determinants of obesity from the perspective of energy economics is relatively limited. We contribute to the literature on the determinants of obesity by empirically studying the relationship between the school energy consumption and children's BMI. Based on a combined dataset of Chinese children's physical health data, kindergarten energy consumption data, and kindergarten geographic information data, we find that school energy consumption is negatively correlated with obesity, and there is considerable heterogeneity in the relationship of school energy consumption between kindergartens in cold areas and severe cold areas and between young girls and young boys. Our results are robust to alternative modeling techniques, the inclusion of additional control variables, and unobservable potential effects. We also find that children's exercise ability is an important transmission channel between school heating and the probability of obesity.

**Keywords:** energy consumption; children's obesity; exercise ability; heating policy

## 1. Introduction

Energy is an important input for heat generation, social production and human survival. In the past forty years, energy has played the role of an engine, bringing to China's economy a world-renowned growth scale and growth rate [1,2]. In addition to economic welfare, energy is also closely related to human development and well-being. Efficient energy consumption can prevent potential health risks, achieve higher educational outcomes, and play an important role in achieving sustainable development [3–6].

Obesity is an important health risk that people face. Some studies have shown that an increase in the obesity rate is not conducive to the realization of the double carbon target [7]. On the one hand, the increased calorie intake by obese and overweight individuals increases food production and food waste [8–10]; on the other hand, increased obesity rates can drive people to choose larger, less energy-efficient vehicles, thereby increasing carbon emissions [8,9,11]. Based on this, linking the incidence of obesity with energy consumption and exploring the health risk aversion effect of energy consumption provide new ideas for understanding energy consumption and the realization of sustainable development goals and dual carbon goals.

Childhood obesity is a serious global health problem. Over the past three decades, the prevalence of obesity and overweight among children worldwide has shown an increasing trend, and it has become an increasingly serious public health problem, similar to adult obesity. Childhood obesity begins in early life in preschoolers, and for many, obesity continues into childhood and into later adulthood. Once identified, obesity is difficult to reverse and is associated with poorer health outcomes in the short and long term. Higher body mass index (BMI) and obesity in childhood are associated with future risk of type 2 diabetes, cardiovascular disease, certain cancers, lack of academic performance,

and mental health problems [12–14]. Furthermore, weight gained during childhood and adolescence is difficult to lose and can lead to overweight and obesity in adults [15]. Addressing childhood obesity is important to reduce lifetime risk and protect health. The main cause of obesity is an imbalance between energy intake and energy expenditure [16]. Existing research has largely identified physical activity, medications, amount of sleep, genetics, stress, and depression as some contributing factors to obesity [17]. Meanwhile, the links between the built, food environment and the natural environment and their relationship to childhood obesity have also been extensively studied.

Previous studies have shown that extreme temperature is an important factor affecting childhood obesity. In the face of temperature shocks, energy use is crucial; energy scarcity is also considered a public health problem both in Europe and internationally. Research has shown that energy poverty is associated with many outcomes, such as health [18–24], education [25,26], and subjective well-being [24,27,28]. Kushneel Prakash et al. [29] concluded, based on Australian panel data research, that household energy deprivation will affect the obesity of family members. School is an important place for children's activities. Thus, the possible association between school energy use and childhood obesity requires consideration. To our knowledge, no previous studies have examined the impact of school energy consumption on childhood obesity. In this paper, we address a critical gap in the literature by examining the relationship between school energy consumption and childhood obesity using cross-sectional data from Chinese children aged 3 to 6 years old. An implicit motivation for such research is that by understanding the role of school energy in influencing body weight, we can improve our understanding of the determinants of obesity and establish targeted intervention policies to curb the obesity crisis.

This paper is based on student data from a 2018 survey conducted by the School of Physical Education of Beijing Normal University as well as National Education Funding Data of the China Education Economic Information Research Center from Beijing Normal University. Our data comes from Beijing Normal University Children's Physical Fitness Monitoring Network, which used a stratified cluster sampling method to sample over 119,000 young children from 1185 kindergartens in 24 provinces of China. Our main indicators for physical fitness included height, weight, BMI, and several physical tests such as the 10 m turn back run, standing long jump, double foot continuous jump, sitting forward flexion, tennis long throw, and walking on the balance beam. We obtained data from these respondents across 94 cities and 24 provinces in China. We measure children's obesity using a continuous variable that equals children's Body Mass Index (BMI) score. This paper measures school energy consumption using the heating cost per student and the heating fee per unit area of every kindergarten to measure school energy consumption. After controlling for student demographic characteristics, school characteristics, natural environment characteristics of districts and counties, and district-county fixed effects, the results of this paper suggest that school energy consumption reduces obesity levels in children; these findings are robust for a range of sensitivity tests. Furthermore, our results are also robust to using school to main coal-producing areas as instrumental variables to address endogeneity issues. In addition, we explore the mechanisms by which school energy consumption affects childhood obesity, and we find that children's exercise abilities comprise the main channel through which school energy consumption affects childhood obesity. Before we conducted this empirical analysis, the following three hypotheses were proposed:

**Hypothesis 1 (H1).** *Holding all other variables constant, there is a significant negative correlation between school energy consumption and BMI among children.*

**Hypothesis 2 (H2).** *There is considerable heterogeneity in the impacts of school energy consumption between kindergartens in cold areas and severe cold areas and between young boys and young girls.*

**Hypothesis 3 (H3).** *Children's athletic abilities are important transmission channels through which school energy consumption influences the probability of being obese.*

This paper uses data from children aged 3–6 in China in 2018 to study the impact of school energy consumption on childhood obesity. There are three main reasons that this study is situated in China. First, China's childhood obesity rate is growing at a high rate. If effective measures are not taken, China's obesity rate may catch up with or exceed that of some developed countries. If no measures are taken, the childhood obesity rate in China is expected to reach 0.6% by 2030, and the number of obese children will increase to 6.64 million [30]; by 2035, the obesity rate for boys will exceed 50%, and the obesity rate for girls will be close to 40% [31]. Second, due to the high obesity rate, China bears a huge burden of disease costs; it is estimated that by 2030, the direct economic cost of adult obesity-related chronic diseases caused by overweight and obesity in China will also increase to 49.05 billion per year [30]. Based on the recent findings by the World Obesity Federation [31], a continuous surge in the annual growth rate of obesity among children and adolescents in China is expected, with a projected increase of 6.6% from 2020 to 2035. Finally, in recent years, school energy consumption has accounted for a larger share of education expenditure. Therefore, using Chinese data to understand the impact of school energy consumption on obesity not only supports policy making in China but also has important policy implications for many developing countries with increasing obesity rates.

The rest of the paper is organized as follows. In Section 2, we discuss our samples, data sources, and research model. Section 3 reports our empirical findings, while Section 4 presents our conclusions and discusses our results.

## 2. Research Design

### 2.1. Sample Selection and Data Sources

This paper selects the physical monitoring data of children in some kindergartens in provinces where central heating is implemented in winter, as well as the financial data of these kindergartens, for research. Considering that there is a time lag in the impact of kindergarten heating expenditure on young children's bodies, this paper selects the 2018 children's physique monitoring results, the kindergarten's financial indicators, and other control variables from the statistical data of 2017. Table 1 shows the number of provinces, schools, and young children involved in the study sample.

**Table 1.** Number of schools and young children in each province.

| Province | Number of Schools | Number of Young Children |
|---|---|---|
| Inner Mongolia | 6 | 1386 |
| Shandong | 74 | 6619 |
| Shanxi | 2 | 105 |
| Xinjiang | 3 | 305 |
| Hebei | 17 | 1818 |
| Henan | 10 | 1240 |
| Liaoning | 4 | 653 |
| Shaanxi | 5 | 739 |
| Qinghai | 2 | 66 |
| Heilongjiang | 1 | 127 |
| Total | 124 | 13,058 |

## 2.2. Variable Definitions

### 2.2.1. Outcome Variable

In this paper, the BMI index of young child *i* in the physical monitoring data is taken as the outcome variable. The BMI index is the ratio of the weight of young child *i* to the square of their height. The BMI index is a proxy variable for better physical fitness of young children. The higher the index value, the higher the obesity level of the young children, and the lower the physical fitness level. This index has the following advantages: (i) The BMI index represents the obesity degree of young children. BMI is an internationally used index that measures individual obesity and has good comparability among different individuals, which is helpful for this study; (ii) BMI has a unified calculation formula, which is convenient for the quantitative analysis of data in this paper.

In fact, many studies use the BMI index to measure the degree of overweight and obesity in children and adolescents [32–36]. For obese children, BMI expressed as a percentage of a specific BMI threshold is an appropriate measurement parameter. Geserick et al. [15], assessing the dynamics of weight gain in a large number of children aged 0–14, indicated that one of the most important determinants of obesity in adolescents and young people is BMI. Similar views are also reflected in Broccoli et al. [37].

### 2.2.2. Input Variable

The core input variable selected in this paper is the per student school energy consumption of kindergartens in 2017, which is obtained from the ratio of the energy consumption of kindergartens to the annual average number of young children. The higher the consumption of energy per student, the stronger the kindergarten's protection for children through heating. The advantages of choosing this indicator as the core explanatory variable are as follows. (i) This variable is the expenditure of a kindergarten on indoor heating; this indicator has a positive correlation with the indoor temperature of kindergarten, which can better represent the indoor temperature level. (ii) This variable is the actual amount of energy consumption, which is easy to quantify.

### 2.2.3. Control Variables

The control variables selected in this paper are shown in Table 2. The variable age represents the age of young children at the time of physical examination; the variable perbudget represents the per capita public budget expenditure of the counties in 2017; the variables temp_M and pre20_20_M represent the average temperature and precipitation of each county in the heating period, respectively; the variable area represents the building area of each kindergarten; the variable personality represents the average tuition of each kindergarten; and the variable heatingarea represents the energy consumption per unit building area of each kindergarten. In this paper, the core explanatory variable heatfee will be replaced by the variable heatingarea in the robustness test for regression analysis. Finally, the variable gender represents the gender of child *i*, with male coded as 1 and female coded as 0.

**Table 2.** Descriptive Statistics.

|  | (1) | (2) | (3) | (4) | (5) |
|---|---|---|---|---|---|
| **VARIABLES** | **N** | **Mean** | **Sd** | **Min** | **Max** |
| age | 13,058 | 4.964 | 0.848 | 3 | 6.920 |
| bmi | 13,058 | 16.11 | 2.104 | 6.805 | 33.21 |
| perbudget | 13,058 | 5074 | 2332 | 2470 | 16,776 |
| energy_con | 13,058 | 385.1 | 543.7 | 4.538 | 4014 |
| temp_m | 13,058 | 5.098 | 3.831 | −8.176 | 9.016 |
| pre20_20_m | 13,058 | 0.585 | 0.243 | 0.203 | 1.803 |
| area | 13,058 | 2574 | 2990 | 254 | 38,428 |
| pertuition | 13,058 | 9272 | 9250 | 0 | 43,105 |
| heatingarea | 13,058 | 24.19 | 18.16 | 0.212 | 127.2 |
| gender | 13,058 | 0.550 | 0.498 | 0 | 1 |

Among the control variables used in this paper, the temperature and precipitation data of each region are from the website of the National Meteorological Administration of China, and the financial indicators of other counties are from the official statistical yearbook.

### 2.3. Research Model

The model setting in this paper is shown in Formula (1):

$$BMI_{isc} = \beta_0 + \beta_1 energy\_con_{is} + X'_{isc}\gamma + u_c + \varepsilon_{isc} \tag{1}$$

$BMI_{isc}$ indicates the *BMI* index of child *i* in kindergarten *s* in county *c*; *energy_con*$_{is}$ represents the school's consumption of energy fees per student of kindergartens; $X'_{isc}$ represents a series of control variables described previously; and we also control the fixed effect of the county $u_c$. The theoretical assumption of this paper is that when the per capita energy consumption is higher, the physical temperature of young children will make them more comfortable, their enthusiasm to participate in classroom activities will be higher, and the obesity level of young children will be lower. Therefore, the coefficient of the core explanatory variable should be significantly negative.

By exploring the relationship between energy poverty and obesity, Kushneel et al. [29] added significant value to the existing body of research on the determinants of obesity. Using data from the Household, Income and Labour Dynamics in Australia (HILDA) survey spanning over 14 years, the authors established a positive association between energy poverty and obesity. Likewise, we observe comparable ideas and methodologies in recent publication [38–40].

## 3. Empirical Results and Analysis

### 3.1. Baseline Regression Analysis

The regression results of Column (3) in Table 3 show that when the control variables are added and the fixed effects at the district and county level are controlled, the estimated coefficient of the core input variables is significantly negative at the level of 1%, which is in line with the theoretical expectation. This regression result shows that increasing indoor temperature has a significant relationship with reducing the obesity level of young children. The results of benchmark regression verify the research hypothesis (H1) of this paper and confirm the research conclusions of Harriet and Carolyn et al. [24].

The baseline regression preliminarily explored the causal relationship between the input variable and the outcome variable, but its internal mechanism remains to be further studied. We will continue to study and analyze several possible influence mechanisms between kindergarten energy consumption and young children's physical fitness in future research after baseline regression.

Following the criteria set by China's National Health Commission ("Growth Standard for Children under 7 Years of Age", WS/T 423-2022), we selected unhealthy groups (inclusive of obese and overweight). We added a dummy variable named "unhealthy" and assigned a value of 1 when a child's weight in our sample exceeded the standard and 0 when within the standard. The outcome variable in our benchmark regression is the dummy variable "unhealthy".

This article uses the logit model (see Formula (2)) to conduct regression, and the results obtained are shown in Table 4. Taking the first column as an example, the estimated coefficient of heating cost per student is negative at the significance level of 1%. Its economic implication is that, with other conditions unchanged, every 100 yuan increase in education cost per student will reduce the average probability of children being unhealthy by 3.64%.

$$Unhealthy_i = \beta_0 + \beta_1 energy\_con_{is} + X'_{isc}\gamma + u_c + \varepsilon_{isc} \tag{2}$$

**Table 3.** Empirical Results I.

| VARIABLES | (1) BMI | (2) BMI | (3) BMI |
|---|---|---|---|
| energy_con | −5.895846 *** | −5.613675 *** | −5.803472 *** |
| | (−7.59) | (−7.25) | (−6.51) |
| age | | 0.045323 * | 0.046518 ** |
| | | (1.93) | (1.97) |
| gender | | 0.368818 *** | 0.369928 *** |
| | | (10.35) | (10.39) |
| area | | | 0.000045 *** |
| | | | (4.15) |
| temp_m | | | −0.315840 *** |
| | | | (−6.38) |
| pre20_20_m | | | 0.035535 |
| | | | (0.26) |
| pertuition | | | −0.040287 |
| | | | (−1.05) |
| perbudget | | | −0.911287 *** |
| | | | (−3.67) |
| constant | 17.268074 *** | 19.066440 *** | 18.431527 *** |
| | (109.15) | (32.48) | (45.79) |
| observations | 13,058 | 13,058 | 13,058 |
| R-squared | 0.058 | 0.065 | 0.067 |
| county FE | YES | YES | YES |

Note: *** $p < 0.01$, ** $p < 0.05$, and * $p < 0.1$; robust t-statistics in parentheses.

**Table 4.** Empirical Results II.

| VARIABLES | (1) Unhealthy | (2) Unhealthy | (3) Unhealthy |
|---|---|---|---|
| heating_fee_per | −0.000364 *** | −0.000404 *** | −0.000339 ** |
| | (−2.61) | (−2.83) | (−2.07) |
| age | | −0.019825 | −0.020886 |
| | | (−0.57) | (−0.60) |
| gender | | 0.024127 | 0.024400 |
| | | (0.43) | (0.44) |
| area | | 0.000026 * | 0.000026 * |
| | | (1.90) | (1.90) |
| temp_m | | | −0.208501 ** |
| | | | (−2.43) |
| pre20_20_m | | | 0.024391 |
| | | | (0.10) |
| pertuition | | | −0.000005 |
| | | | (−0.86) |
| perbudget | | | −0.000075 |
| | | | (−1.61) |
| constant | −1.432469 *** | −1.370255 *** | −0.229387 |
| | (−4.99) | (−4.12) | (−0.31) |
| observations | 13,040 | 13,040 | 13,040 |
| county FE | YES | YES | YES |

Note: Robust z-statistics in parentheses. *** $p < 0.01$, ** $p < 0.05$, and * $p < 0.1$. The outputs of the model in this paper are all mean marginal effects.

From this regression, we discovered that winter heating expenditures for kindergartens have a significant, negative correlation with the probability of being overweight or obese.

### 3.2. Heterogeneity Analysis

Considering that the climate in different regions will have different impacts on the heating standards and physical fitness of young children, this paper refers to the winter indoor heating standards issued by the Ministry of Housing and Urban Rural Development of China and divides the regions involved in the sample into cold regions and severe cold regions (with reference to the official standard of the Ministry of Housing and Urban Rural Development of China (Code for Thermal Design of Civil Buildings (GB50176–2016)), cold regions include Liaodong Peninsula, North China, and parts of the Loess Plateau, while severe cold areas include Heilongjiang, Jilin, northern Liaoning, Inner Mongolia, Xinjiang, and Qinghai). The specific division standard is that if the minimum daily average temperature of the local area over the years is less than or equal to $-10$ °C, the area is recognized as a severe cold area; according to the requirements of the Ministry of Housing and Urban Rural Development, indoor heating in winter should be given priority to civil buildings in severe cold areas, and the relevant construction standards are higher than those in other regions of China. If the minimum daily average temperature of the local area over the years is greater than $-10$ °C and less than or equal to 0 °C, the area is considered a cold area. The heating standard of civil buildings in cold areas is slightly lower than that in severely cold areas. Based on this feature, this paper will analyze the heterogeneity of the sample in a subsequent empirical study.

Figure 1 shows the regional distribution of China's centralized heating policy in winter. Among these areas, the light blue part represents the cold areas involved in this paper, and the dark blue part represents the severe cold areas. The number of provinces in the figure is the sample number used in this paper.

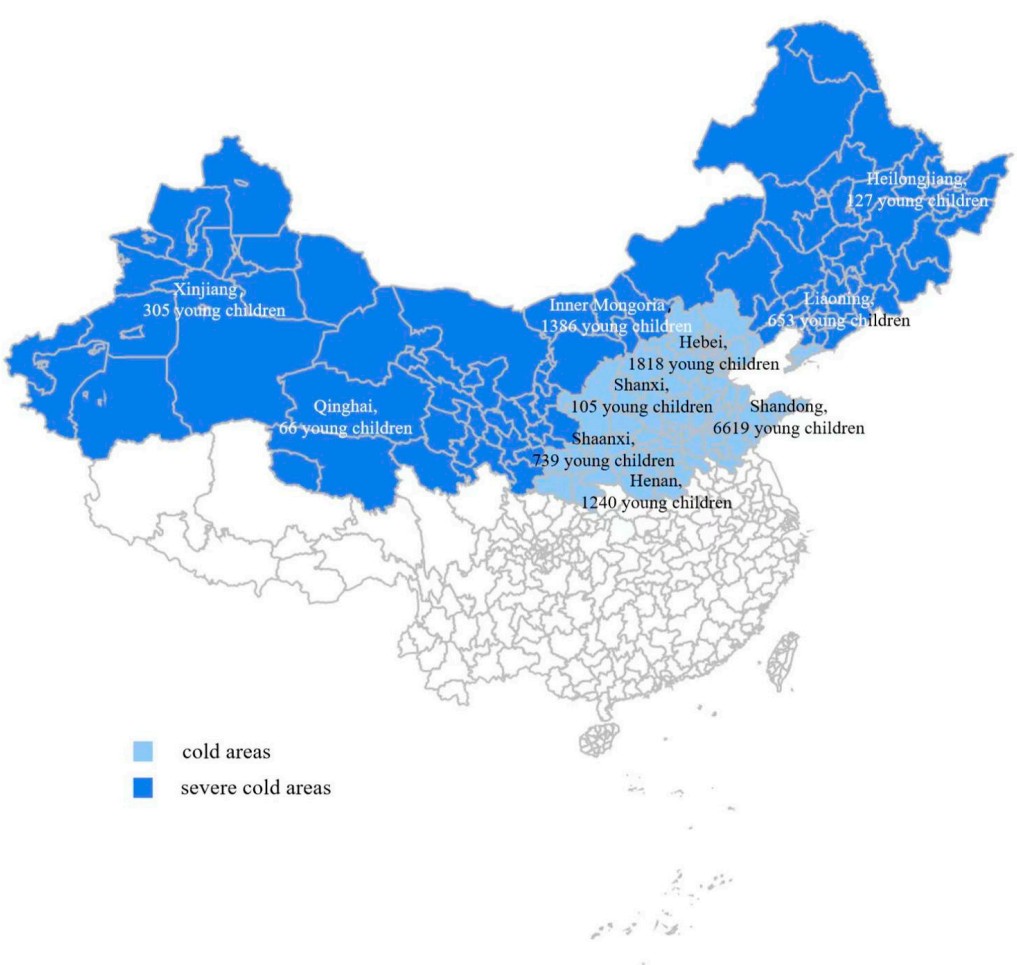

**Figure 1.** Division of cold area and severe cold area and sample quantity of each province.

As shown in Column (1) and Column (2) of Table 5, the regression results of the samples from cold areas and severely cold areas show that the coefficients of core explanatory variables are significantly negative at the levels of 1% and 5%, respectively, which is in line with the theoretical expectation of this paper. That is, in different climatic regions, the relationship between winter energy consumption of kindergartens and childhood obesity is different, and the energy consumption of kindergartens in cold regions has higher marginal benefits. According to a study by Charles J. Vierck et al. [41], females display a greater sensitivity than males to brief nociceptive stimuli, such as heat and cold. Given the potential for varying sensitivities to temperature among children of different genders, this study conducts an analysis to explore heterogeneity according to gender. The regression results of Columns (3) and (4) show that when the average heating cost of kindergarten students increases, the BMI index of boys and girls decreases significantly; at the same time, the absolute value of the core input variable coefficient in the sample of young boys is higher than that of young girls, indicating that young boys are more affected by the marginal impact of heating costs. The regression results of the heterogeneity analysis verify the second research hypothesis (H2) of this paper.

**Table 5.** Heterogeneity Analysis Results I.

| VARIABLES | (1) Cold Areas BMI | (2) Severe Cold Areas BMI | (3) Young Boys BMI | (4) Young Girls BMI |
|---|---|---|---|---|
| energy_con | −5.910157 *** | −74.001988 ** | −6.231438 *** | −5.232211 *** |
|  | (−6.57) | (−2.17) | (−5.31) | (−3.81) |
| age | 0.084463 *** | −0.098022 | 0.127587 *** | −0.048705 |
|  | (3.37) | (−1.44) | (3.85) | (−1.46) |
| gender | 0.360687 *** | 0.405058 *** |  |  |
|  | (9.15) | (4.46) |  |  |
| area | 0.000045 *** | 0.007017 *** | 0.000049 *** | 0.000040 ** |
|  | (4.19) | (2.96) | (3.66) | (2.34) |
| temp_m | 4.646316 *** | 101.771957 ** | −0.271049 *** | −0.376241 *** |
|  | (6.47) | (2.57) | (−3.95) | (−5.29) |
| pre20_20_m | 60.347691 *** | 1751.791977 ** | 0.014838 | 0.109248 |
|  | (7.19) | (2.57) | (0.08) | (0.58) |
| pertuition | −0.039021 | −5.411726 *** | −0.014955 | −0.078753 |
|  | (−1.00) | (−2.76) | (−0.29) | (−1.38) |
| perbudget | 21.237331 *** | −424.371647 ** | −0.529649 | −1.319732 *** |
|  | (6.71) | (−2.55) | (−1.48) | (−3.84) |
| constant | −62.748030 *** | 66.600307 *** | 17.982553 *** | 19.382346 *** |
|  | (−5.50) | (2.92) | (31.78) | (34.12) |
| observations | 10,742 | 2045 | 7181 | 5877 |
| R-squared | 0.072 | 0.035 | 0.057 | 0.072 |
| county FE | YES | YES | YES | YES |

Note: *** $p < 0.01$, ** $p < 0.05$; robust t-statistics in parentheses.

As shown in Column (1) and Column (2) of Table 6, when the average energy consumption of kindergarten students increases, the obesity of boys and girls in cold areas decreases, which is in line with theoretical expectations. Columns (3) and (4) of Table 6 show the extent to which the BMI of boys and girls in severely cold areas has a significantly negative relationship with the average energy consumption of kindergarten students. The regression results of the sample of girls and young children in severely cold areas are in line with expectations, while the obesity level of boys and young children in severely cold areas have no significant relationship with the average energy consumption of kindergarten students. The temperature in cold areas is higher in winter, which is suitable for sports activities. The average energy consumption of students has an obvious relationship with the physical fitness of young children.

**Table 6.** Heterogeneity Analysis Results II.

| | (1) | (2) | (3) | (4) |
|---|---|---|---|---|
| | **Young Boys in Cold Areas** | **Young Girls in Cold Areas** | **Young Boys in Severe Cold Areas** | **Young Girls in Severe Cold Areas** |
| **VARIABLES** | **BMI** | **BMI** | **BMI** | **BMI** |
| energy_con | −6.299038 *** | −5.351836 *** | −64.504882 | −99.609391 * |
| | (−5.33) | (−3.85) | (−1.46) | (−1.83) |
| age | 0.182652 *** | −0.032886 | −0.110302 | −0.074809 |
| | (5.24) | (−0.91) | (−1.15) | (−0.81) |
| area | 0.000050 *** | 0.000041 ** | 0.007020 ** | 0.007490 ** |
| | (3.69) | (2.38) | (2.21) | (2.07) |
| temp_m | 3.917841 *** | 5.14806 8*** | 94.299718 * | 124.016242 ** |
| | (4.00) | (4.97) | (1.81) | (2.00) |
| pre20_20_m | 51.002417 *** | 67.181079 *** | 1623.949365 * | 2133.196409 ** |
| | (4.46) | (5.55) | (1.81) | (2.00) |
| pertuition | −0.016556 | −0.075871 | −5.058008 * | −6.298797 ** |
| | (−0.32) | (−1.30) | (−1.93) | (−2.10) |
| perbudget | 18.173904 *** | 23.327238 *** | −391.848599 * | −519.914738 ** |
| | (4.20) | (5.12) | (−1.79) | (−2.00) |
| constant | −50.718840 *** | −70.814425 *** | 60.093446 ** | 83.938330 ** |
| | (−3.26) | (−4.30) | (2.03) | (2.31) |
| observations | 5957 | 4785 | 1088 | 957 |
| R-squared | 0.063 | 0.079 | 0.034 | 0.026 |
| county FE | YES | YES | YES | YES |

Note: *** $p < 0.01$, ** $p < 0.05$, and * $p < 0.1$; robust t-statistics in parentheses.

*3.3. Robustness Test*

Considering the high correlation between the energy consumption per student and the energy consumption per unit building area in the school, this paper uses the variable heatingarea, which can represent the heating cost per unit building area of kindergarten, to replace the original core explanatory variable for analysis. Table 7 shows the regression results of the robustness test for the whole sample, cold area, and severely cold area. In the three column regression results, the regression coefficients of the core input variables are significant at the level of 1%, which conforms to the theoretical expectation and confirms the research hypothesis of this paper.

At the same time, this paper also notes that the absolute value of the regression coefficient of the core input variable of the sample in the cold region is higher than the absolute value of the regression coefficient in the cold region. This result also echoes the second research hypothesis (H2) of this paper; that is, the average energy consumption of kindergartens in different regions has a different relationship with the physical fitness of young children.

Table 8 shows the robustness test results based on the two dimensions of region and young children's gender. Among them, Columns (1) and (2) show the regression results of samples of boys and girls in cold areas. The coefficients of the core explanatory variables in the two regression results are significantly negative, which is in line with the expectations of the robustness test. Columns (3) and (4) are the regression results of samples of boys and girls in severely cold areas. The results show that the regression coefficients of the core explanatory variables are significantly negative at the level of 10% for the regression of boys and young children in severely cold areas; the fourth table shows that the regression coefficient of the core explanatory variable is significantly negative at the level of 1% in the regression of the sample of girls and young children in severely cold areas. The above results are in line with the theoretical expectation; that is, the average energy consumption of kindergarten students is associated with the health level of young children, which passed the robustness test. The regression results shown in Table 8 are consistent with the research

conclusions of Somerville et al. [23]. The latter found that the incidence rate of children with asthma was significantly reduced after providing central heating systems for families.

**Table 7.** Robustness Test Results I.

|  | (1) | (2) | (3) |
|---|---|---|---|
|  | All | Cold Areas | Severe Cold Areas |
| VARIABLES | BMI | BMI | BMI |
| heatingarea | −148.046020 *** | −150.230756 *** | −2520.156454 *** |
|  | (−7.29) | (−7.28) | (−3.24) |
| age | 0.052669 ** | 0.090551 *** | −0.091934 |
|  | (2.23) | (3.60) | (−1.37) |
| gender | 0.373448 *** | 0.364334 *** | 0.402858 *** |
|  | (10.49) | (9.25) | (4.46) |
| area | 0.000044 *** | 0.000045 *** | −0.000341 |
|  | (4.16) | (4.21) | (−0.32) |
| temp_m | −0.436481 *** | 3.803279 *** | −18.012537 * |
|  | (−8.31) | (5.14) | (−1.66) |
| pre20_20_m | 0.234674 * | 51.819954 *** | −326.052160 * |
|  | (1.70) | (6.03) | (−1.71) |
| pertuition | −0.254972 *** | −0.265286 *** | 4.356076 ** |
|  | (−7.08) | (−7.22) | (2.57) |
| perbudget | −1.373121 *** | 17.557089 *** | 75.025837 * |
|  | (−5.46) | (5.40) | (1.70) |
| constant | 19.468913 *** | −49.949912 *** | 13.358363 *** |
|  | (45.48) | (−4.26) | (6.93) |
| observations | 13,058 | 10,742 | 2045 |
| R-squared | 0.068 | 0.073 | 0.041 |
| county FE | YES | YES | YES |

Note: *** $p < 0.01$, ** $p < 0.05$, and * $p < 0.1$; robust t-statistics in parentheses.

**Table 8.** Robustness Test Results II.

|  | (1) | (2) | (3) | (4) |
|---|---|---|---|---|
|  | Young Boys in Cold Areas | Young Girls in Cold Areas | Young Boys in Severe Cold Areas | Young Girls in Severe Cold Areas |
| VARIABLES | BMI | BMI | BMI | BMI |
| heatingarea | −141.827338 *** | −157.754650 *** | −2029.257005 * | −3472.569564 *** |
|  | (−4.98) | (−5.28) | (−1.91) | (−2.92) |
| age | 0.187460 *** | −0.024843 | −0.109234 | −0.059445 |
|  | (5.38) | (−0.69) | (−1.15) | (−0.66) |
| area | 0.000049 *** | 0.000041 ** | 0.000992 | −0.002867 |
|  | (3.67) | (2.41) | (0.83) | (−1.45) |
| temp_m | 3.184990 *** | 4.185210 *** | −6.396495 | −40.720527 ** |
|  | (3.15) | (3.93) | (−0.50) | (−2.12) |
| pre20_20_m | 43.706363 *** | 57.292328 *** | −121.931445 | −724.733338 ** |
|  | (3.72) | (4.63) | (−0.54) | (−2.14) |
| pertuition | −0.244703 *** | −0.295864 *** | 2.951146 | 7.230641 ** |
|  | (−4.97) | (−5.34) | (1.37) | (2.56) |
| perbudget | 14.956716 *** | 19.143664 *** | 28.451525 | 166.288260 ** |
|  | (3.36) | (4.09) | (0.54) | (2.13) |
| constant | −39.642254 ** | −56.130190 *** | 14.024249 *** | 12.077910 *** |
|  | (−2.47) | (−3.32) | (5.18) | (4.29) |
| observations | 5957 | 4785 | 1088 | 957 |
| R-squared | 0.063 | 0.081 | 0.037 | 0.037 |
| county FE | YES | YES | YES | YES |

Note: *** $p < 0.01$, ** $p < 0.05$, and * $p < 0.1$; robust t-statistics in parentheses.

Comparing the regression results of Column (3) and Column (4), it can be found that the marginal influence of the average energy consumption of kindergarten students on the physical obesity of young girls in severely cold areas is greater. This shows that there is a certain gender difference in the influence of the average energy consumption of kindergarten students on the physical quality of young children in severely cold areas. This robustness test result echoes the research of Jianxiong Hu et al. [19]. The latter found that the central heating policy significantly reduced winter mortality in northern China. Moreover, this policy has a more significant positive effect on the health of women and elderly individuals.

### 3.4. Mechanism Analysis

This paper argues that young children's participation in classroom sports activities is an important intermediary variable. In this paper, the shuttle run, the double jump, and the balance beam are taken as the explained variables for mechanism analysis.

The regression results in Table 9 show that the results in Column (1) and Column (2) are in line with the theoretical expectation of this paper; that is, when the average heating expenditure for kindergarten students increases, the performance of children in the shuttle run and double jump will be better. The results of this paper are similar to Yukie Hayashi et al. [42]'s conclusion that indoor temperature affects the physical function of elderly individuals. The difference is that the research object of this paper is preschool children, which improves and supplements the research perspective and conclusions of the previously mentioned article.

**Table 9.** Mechanism Analysis Results.

| VARIABLES | (1)<br>Shuttle Run | (2)<br>Jump_Two | (3)<br>Balance_Beam |
|---|---|---|---|
| energy_con | −1.879702 ** | 2.823611 * | −2.541689 |
| | (−2.41) | (1.67) | (−0.61) |
| age | −1.214818 *** | −1.932578 *** | −5.850008 *** |
| | (−60.48) | (−50.19) | (−50.75) |
| gender | −0.194671 *** | −0.010330 | 0.201678 |
| | (−7.16) | (−0.19) | (1.23) |
| area | −0.000002 | 0.000032 *** | 0.000045 |
| | (−0.37) | (3.30) | (1.41) |
| temp_m | −0.093344 | −0.248471 ** | −1.399235 *** |
| | (−1.41) | (−2.20) | (−3.51) |
| pre20_20_m | −0.832104 *** | −0.679561 ** | 2.367412 *** |
| | (−5.90) | (−2.46) | (2.88) |
| pertuition | −0.250570 *** | −0.275795 *** | 0.225424 |
| | (−7.30) | (−4.11) | (1.19) |
| perbudget | −1.450105 *** | −2.168258 *** | −5.745470 *** |
| | (−5.17) | (−4.14) | (−3.05) |
| constant | 16.953163 *** | 21.190718 *** | 53.436628 *** |
| | (31.50) | (22.91) | (15.98) |
| observations | 12,880 | 12,876 | 12,800 |
| R-squared | 0.359 | 0.260 | 0.285 |
| county FE | YES | YES | YES |

Note: *** $p < 0.01$, ** $p < 0.05$, and * $p < 0.1$; robust t-statistics in parentheses.

### 3.5. Analysis of Endogeneity Problems

The model in this paper may omit variables; thus, it is necessary to use the variable "driving distance from each kindergarten to the nearest main coal production area" as an instrumental variable to solve the endogeneity problem. The driving distance from the kindergarten to the nearest coal production area is highly correlated with the energy consumption of the kindergarten, meeting the correlation requirements of tool variables; however, this variable cannot affect the physical quality of young children through other

channels to meet the exclusive requirements. Columns (1) and (2) of Table 10 show the analysis results in the two-stage regression. The regression result of the first stage is significantly negative. The farther away from the main coal production area that the school is, the lower the school's coal consumption expenditure; the result of the two-stage regression is significantly negative, which is in line with expectations and verifies the theoretical hypothesis of this paper.

**Table 10.** Regression Results of Instrumental Variables.

| | (1) | (2) |
|---|---|---|
| | **Energy_Con** | **BMI** |
| coaldist | −0.333 *** | |
| | (−5.66) | |
| energy_con | | −0.003 *** |
| | | (−3.57) |
| age | −21.869 *** | 0.001 |
| | (−5.91) | (0.02) |
| gender | −5.615 | 0.382 *** |
| | (−0.84) | (9.08) |
| area | 0.006 *** | 0.000 *** |
| | (8.66) | (5.00) |
| temp_m | −22.199 *** | −0.042 ** |
| | (−17.31) | (−2.04) |
| pre20_20_m | −89.340 *** | −0.321 *** |
| | (−10.25) | (−2.99) |
| pertuition | 0.041 *** | 0.000 *** |
| | (46.61) | (3.60) |
| perbudget | 0.037 *** | 0.000 *** |
| | (21.54) | (4.13) |
| _cons | 127.963 *** | 15.511 *** |
| | (5.71) | (88.38) |
| N | 13036 | 13036 |
| r2_a | 0.516 | |
| F | 570.57 | |

Note: *** $p < 0.01$, ** $p < 0.05$; robust t-statistics in parentheses.

## 4. Discussion and Conclusions

### 4.1. Research Conclusions

This paper analyzes the relationship between energy consumption in winter kindergartens and the physical fitness of young children by using the monitoring data of young children's physique, the financial data of kindergartens, and other control variables. Through the above mentioned research, this paper draws the following conclusions.

First, when the energy consumption of kindergartens increases, the obesity level of young children decreases. The possible internal mechanism is that the increase in heating expenditure makes the room temperature rise, which guarantees the willingness and enthusiasm of children to participate in sports activities. When the frequency of children's participation in sports activities increases, their own obesity will also decrease. This finding is consistent with Sefa et al. [28]'s research results and refines Lei Pan et al. [5]'s research on the relationship between energy poverty and the average public health level. It also confirms Ziyu Zhang et al. [43]'s research on the relationship between multidimensional poverty and residents' health from the perspective of individual obesity levels. In addition, the research conclusions based on school level data also supplement and improve the research results on the relationship between energy poverty and health obtained by Dayong Zhang et al. [26] from household-level data.

Second, through the analysis of heterogeneity, it is found that for cold areas, the energy consumption of kindergartens significantly reduces the obesity level of boys and girls; however, for severely cold areas, the heating cost of kindergartens has a significant

relationship with girls' obesity reduction, while less conclusive results were found for the regression of the boys' samples. The internal reason may be that because of the cold winter climate, schools generally do not carry out sports activities; thus, the marginal impact of energy consumption on the physical fitness of young children is weak, which can be confirmed in the regression results of boys and young children in severely cold areas. Kindergarten energy consumption has different effects on children of different regions and genders. The internal mechanism of this phenomenon needs to be further explored in a follow-up study. The research results of this paper on the correlation between energy consumption and the physical fitness of young boys and girls echo Shamal et al. [20]'s research on energy poverty in early childhood development.

Third, this paper uses the heating cost per unit building area of kindergarten to replace the original core input variable for robustness test. The results confirm the research hypothesis of this paper, which is consistent with the research conclusion of Howden Chapman et al. [44], who reported that low housing construction and heating standards have had a negative impact on the health of some New Zealand residents.

Finally, this paper finds that when kindergartens increase energy consumption, young children's willingness to participate in school sports activities increases, as specifically seen in the children's improved performance in the ten-meter round-trip run and the double jump and thus lower obesity levels. This research finding confirms the research conclusion of Kushneel et al. [29], who found that energy poverty can indeed cause an increase in individual obesity.

### 4.2. Practical Insights

Based on the analysis of this paper, we can draw the following suggestions. (i) We should pay attention to the impact of the school learning environment on the physical attributes of young children. The results of this study show that energy consumption in kindergartens has a significant positive effect in reducing children's obesity. To achieve a better expected output of preschool education, kindergartens should focus their education expenditure on energy consumption and other consumption items that directly affect the living and learning environment of young children. (ii) Attention should be given to the differences in the impact of energy consumption on young children of different regions and genders. Girls in severely cold areas are more affected by heating funds, while boys are not significantly affected. In the future, we should pay attention to the differences in the physical quality of children of different genders to achieve balanced development among different groups. (iii) Public funding should address energy poverty at the school level. Combining the research results of this paper and previous studies, it can be seen that the energy consumption of kindergartens can play an important role in the physical and mental development of young children through different channels. Therefore, public funding should aim to reduce the occurrence of energy poverty at the school level to ensure the healthy physical and mental development of young children.

### 4.3. Research Contribution

This article makes an important contribution to the literature in three main areas. First, this paper contributes to the general literature on school energy consumption and the impact of energy consumption in schools [45,46]. While a body of literature examines school energy consumption patterns and the impact of school energy consumption on learning, there is no literature examining the impact of energy consumption on childhood obesity. This article adds to this literature by examining the impact of school energy consumption on obesity. Second, we supplement the literature on drivers of obesity [47–51]. We find that school energy consumption has a significant relationship with obesity in addition to established factors, including physical activity, medications, sleep amount, genetics, stress, depression, and poor diet. Finally, we add to the limited literature on the channels that may account for the impact of school energy consumption on children's obesity. A small body of literature has examined how the amount of sleep, health status, and

psychological distress affect children's obesity [29]. We find that school energy consumption levels can significantly affect children's athletic ability, which is important for lowering obesity levels. Vanhelst et al. [52] reported that exercise intervention programs geared towards obese children and youth can precipitate a decrease in BMI and lead to improved athletic performance. Additionally, Ahmed et al. [53] discovered certain risk factors for obesity/overweight, including aspects such as male gender, middle socioeconomic status, poor self-rated athletic ability, attempts to lose weight, and insufficient consumption of fruit (<4 times a week).

### 4.4. Research Limitations

The biggest limitation of this paper is that it was not possible to obtain information about young children at the family level and the parent level; furthermore, it was not possible to control the influencing factors of families and parents in the model, which leads to endogeneity problems. To solve this problem, this study uses constructed variables. At the same time, in the mechanism analysis section, we only discuss the impact of energy consumption on the physical test results of young children and do not consider other possible mechanisms.

**Author Contributions:** Conceptualization, Y.Z. and S.H.; methodology, S.H. and B.H.; project administration, S.H.; resources, Y.Z. and L.F.; software, S.H. and B.H.; writing—original draft preparation, S.H. and B.H.; writing—review and editing, Y.Z. and B.H.; supervision, L.F. and X.L. All authors have read and agreed to the published version of the manuscript.

**Funding:** This research was funded by the National Social Science Foundation under Award Number 22FTJB001 (Research on the allocation of educational resources for the construction of an educational power).

**Institutional Review Board Statement:** The study was conducted according to the guidelines of the Declaration of Helsinki, and approved by the Institutional Review Board of the Faculty of Psychology, BNU (IRB Number: 201903260036; 2014–2019).

**Informed Consent Statement:** Informed consent was obtained from all subjects involved in the study.

**Data Availability Statement:** The data are not publicly available due to ethical reasons.

**Conflicts of Interest:** The authors declare no conflict of interest.

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
