# Peer review of "School Energy Consumption and Children’s Obesity: Evidence from China"

_sustainability, doi:10.3390/su15108226_

Round 1

Reviewer 1 Report

The work presented is very interesting and contains important research, but the manuscript itself needs several corrections. 

1/Presented study is a correlational study. From a correlational study we cannot infer causality but only co-occurrence. Unfortunately, the authors repeatedly write about the effect of energy consumption on obesity, which is a major methodological error. We can only speak of co-occurrence of these variables. 

E.g. Line 69-70
In this paper, we fill an important gap in the literature by using cross-sectional 69 data from children aged 3-6 years in China to examine how school energy consumption 70 affects obesity. 

Statements of impact even appear in hypotheses 
Hypothesis 1 (H1). All else being equal, school energy consumption significantly inhibits children's BMI. 

2/ BMI is used as the dependent variable in this study. This very popular indicator is unfortunately not ideal. Especially in children, the diagnosis of obesity based on BMI is made with reference to centile grids developed for the population. In the discussion of the results, the authors write about the influence of energy consumption on obesity ... in my opinion, this is too far-reaching a generalisation. From the results of the study we can infer the co-occurrence of energy consumption and BMI but probably not the effect on obesity .... 

3/ There is too little information in the manuscript about the measurement procedure of the children. How were the variables measured (height/weight) from which BMI was obtained ? Were all children tested in the same way and when were these measurements taken ?

4/ Including centile measures, it would be useful to supplement the data with information on how many of the children surveyed were overweight and how many were obese ?

Reviewer 2 Report

I have a hard time understanding the rationale behind this study

Authors wanted to assess the relantionship between obesity and energy consumption, but school energy consumption can be influenced by many factors, rather than by student's BMI directly.

Line 91. What do you mean by "inhibits children's BMI?" I don't see the association between body mass index and school energy consumption rates

Line 104. Isn't there a more recent source? It could be updated

Line 112. Do not state personal opinions in the Introduction. These statements should be reserved for the Discussion section

Lines 126-128. How? This is a very bold statement, this should be referenced, if possible

Line 142. I don't think variables are properly established.

There is no "dependent" or "independent" variable in a correlational analysis, as this paper intends to do.

These are established when an intervention is applied, and the effect on this intervention is assessed over a variable that changes

Line 144. I suggest using the term "outcome" instead of "variable"

Line 144-152. You could use a reference that supports this information, at least for some of the statements that you make

Line 181. Reference for this formula?

Is this formula used based on previous developed research?

Line 210. Into "cold regions" and "cold regions"?

Lines 215-216. So under 10º is a severe cold area, but 0º is simply a cold area? I'm not sure if this is the case, either way the way it is explained is confusing. Please rephrase.

Line 223. When stating "the figure" please add the number of the figure

Line 233-234. This should be referenced, if it is something that has been proven as so

Line 250-251. Do not state personal opinions in the Results section, this is reserved for the Discussion.

Line 267-269. Again, comparison with other study results should be reserved for the DIscussion. Please elaborate this information in this section 

Round 2

Reviewer 1 Report

manuscript corrected

good luck with your further scientific work